# *Drosophila* seminal sex peptide associates with rival as well as own sperm, providing SP function in polyandrous females

Snigdha Misra, Mariana F Wolfner*

Department of Molecular Biology and Genetics, Cornell University, Ithaca, New York, United States

**Abstract** When females mate with more than one male, the males' paternity share is affected by biases in sperm use. These competitive interactions occur while female and male molecules and cells work interdependently to optimize fertility, including modifying the female's physiology through interactions with male seminal fluid proteins (SFPs). Some modifications persist, indirectly benefiting later males. Indeed, rival males tailor their ejaculates accordingly. Here, we show that SFPs from one male can directly benefit a rival's sperm. We report that Sex Peptide (SP) that a female *Drosophila* receives from a male can bind sperm that she had stored from a previous male, and rescue the sperm utilization and fertility defects of an SP-deficient first-male. Other seminal proteins received in the first mating 'primed' the sperm (or the female) for this binding. Thus, SP from one male can directly benefit another, making SP a key molecule in inter-ejaculate interaction.

*For correspondence:
mfw5@cornell.edu

## Introduction

In many animal species, females mate with more than one male. This polyandry lays the foundation for differential fertilization success of sperm from the different males (*Parker, 1970*; *Parker, 1979*) within a female, whose 'cryptic female choice' can bias the relative use of these sperm (*Eberhard, 1996*). This, in turn can drive the evolution of male reproductive traits including optimal sperm numbers, morphology, and seminal protein sequences (*Almeida and Desalle, 2008*; *Birkhead, 1998*; *Pitnick and Miller, 2000*).

Against the backdrop of these conflicts, male and female molecules and/or cells must also work together to ensure reproductive success. How efficiently sperm interact with the egg and instigate successful fertilization or embryo support (where relevant) is key to successful fertility. Accordingly, males have evolved molecular mechanisms that trigger physiological changes in females that increase the reproductive success of the mating pair. Seminal fluid proteins (SFPs) are crucial regulators of these changes. SFPs are produced within glandular tissues in the male reproductive tract and are transferred to females along with sperm during mating (*Avila et al., 2010*; *Poiani, 2006*; *Ravi Ram and Wolfner, 2007b*; *Ravi Ram et al., 2005*; *Ravi Ram and Wolfner, 2009*; *Wigby et al., 2009*). Within a mated female, SFPs mediate an array of post-mating responses such as, in insects, changes in egg production, elevated feeding rates, higher activity or reduced sleep levels, long-term memory, activation of the immune system and reduced sexual receptivity (*Avila et al., 2011*; *Bath et al., 2017*; *Scheunemann et al., 2019*; *Isaac et al., 2010*; *Domanitskaya et al., 2007*; *Chapman et al., 2003*; *Schwenke et al., 2016*).

The ability of a male's SFPs to induce long-term changes in the mated female enhances that male's reproductive success. For example, the seminal Sex Peptide (SP) of male *Drosophila* binds to his sperm stored in the female, persisting there for approximately 10 days (*Peng et al., 2005*). This binding of SP to sperm is aided by the action of a network of other SFPs, the 'LTR-SFPs' (*Ravi Ram and Wolfner, 2009*; *Singh et al., 2018*; *Findlay et al., 2014*). The active region of SP is then

**eLife digest** When fruit flies and other animals reproduce, a compatible male and a female mate, allowing sperm from the male to swim to and fuse with the female's egg cells. The males also produce proteins known as seminal proteins that travel with the sperm. These proteins increase the likelihood of sperm meeting an egg and induce changes in the female that increase the number, or quality, of offspring produced.

Some seminal proteins help a male to compete against its rivals by decreasing their chances to fertilize eggs. However, since many of the changes seminal proteins induce in females are long-lasting, it is possible that a subsequent male may actually benefit indirectly from the effects of a prior male's seminal proteins. It remains unclear whether the seminal proteins of one male are also able to directly interact with and help the sperm of another male.

Male fruit flies make a seminal protein known as sex peptide. Normally, a sex peptide binds to the sperm it accompanies into the female, increasing the female's fertility and preventing her from mating again with a different male. To test whether the sex peptide from one male can bind to and help a rival male's sperm, Misra and Wolfner mated female fruit flies with different combinations of males that did, or did not, produce the sex peptide.

The experiments found that female flies that only mated with mutant males lacking the sex peptide produced fewer offspring than if they had mated with a 'normal' male. However, in females that mated with a mutant male followed by another male who provided the sex peptide, the second male's sex peptide was able to bind to the mutant male's sperm (as well as to his own). This in turn allowed the mutant male's sperm to be efficiently used to sire offspring, at levels comparable to a normal male providing the sex peptide.

These findings demonstrate that the ways individual male fruit flies interact during reproduction are more complex than just simple rivalry. Since humans and other animals also produce seminal proteins comparable to those of fruit flies, this work may aid future advances in human fertility treatments and strategies to control the fertility of livestock and pests, including mosquitoes that transmit diseases.

gradually cleaved from sperm in storage, dosing the females to maintain high rates of egg laying, decreased receptivity to remating (*Peng et al., 2005*), increased food intake, and slower intestinal transit of the digested food to facilitate maximum absorption and production of concentrated faeces (*Avila et al., 2011*; *Apger-McGlaughon and Wolfner, 2013*; *Carvalho et al., 2006*; *Gioti et al., 2012*; *Cognigni et al., 2011*). However, induction of these changes can also indirectly benefit his rival, as the female's physiology will have already been primed for reproduction by her first mate's SFPs. Such indirect benefits to the second male have been suggested to explain the tailoring of the ejaculate by males that mate with previously mated females (*Wigby et al., 2009*; *Garbaczewska et al., 2013*; *Sirot et al., 2011*; *Neubaum and Wolfner, 1999*). For example, the *Drosophila* seminal protein ovulin increases the number of synapses that the female's Tdc2 (octopa-minergic) neurons make on the musculature of the oviduct above the amount seen in unmated females (*Rubinstein and Wolfner, 2013*). This is thought to sustain high octopaminergic (OA) signal-ing on the oviduct musculature of mated female, allowing increased ovulation to persist in mated female, even after ovulin is no longer detectable in the female. Therefore, males mating with previ-ously mated females need transfer less ovulin than males mated to virgin females, presumably because it may be less necessary, as they benefit from the ovulation stimulating effect of ovulin from the prior mating. In another example, prior receipt of Acp36DE can rescue sperm storage of a male that lacks this SFP (*Avila and Wolfner, 2009*; *Chapman et al., 2000*).

The benefits to the second male described above are indirect consequences of the first male's SFPs' effects on female's physiology. The second male is thus the lucky beneficiary of the first male's SFPs' actions. However, it is unknown whether a male could directly benefit from a rival's SFPs, for example, whether the latter could associate with and improve the success of another male's sperm. There was some suggestion that this might occur from the phenomenon of 'copulation complemen-tation' (*Xue and Noll, 2000*), in which a female *Drosophila* singly-mated to a male lacking SFPs did not produce progeny unless she remated to a male who provided SFPs. That finding suggested that

something from the second mating allowed the first male's sperm to be used. However, the molecular basis for this phenomenon was unknown. The relevance of such 'complementation' to male reproductive fitness was strengthened by several sperm competition studies, that suggested that a male's reproductive success could benefit from a rival's SFPs. For example, *Avila et al., 2010* reported that the sperm of *SP*-null males were better at defensive sperm competition than the sperm of control males. Specifically, females mated to *SP*-null males and then subsequently remated to a wildtype (wt) competitor produced significantly more progeny from the first male (P1) relative to the P1 of control males who had mated to females before the wt competitor. The higher P1 of *SP*-null males in this situation likely occurred because at the time of the second mating, the mates of *SP*-null males contained more of his sperm compared to the sperm retained from control males. This is because SP is required for efficient release of sperm from storage (*Avila et al., 2010*). The higher P1 of SP-null males suggested that SP received from a second male might promote release of both his sperm and of the stored sperm from the previous SP-null male. However, this had never been tested.

Here, we report that *Drosophila* SP received from a second male can bind to a prior male's SP-deficient sperm and restore his fertility, including sperm release from storage and changes in the female's behavior. We also show that although LTR-SFPs are normally required for SP to bind sperm, sperm from an SP-deficient mating can bind SP from a subsequent male, even if he lacks LTR-SFPs. This suggests that the LTR-SFPs from the first mating 'primed' the sperm (or the female), allowing sperm-binding by subsequently-received SP. Our results reveal direct benefits that previously stored sperm from the first (or prior) male can receive from the second (or last) male's ejaculate during the course of successive matings. Our results also establish SP as a crucial long-term molecule that facilitates this inter-ejaculate interaction, and SP-sperm binding as the molecular mechanism that underlies the reported 'copulation complementation'(*Xue and Noll, 2000*) in *Drosophila*.

## Results

### Sex peptide from one male can associate with sperm from another

In matings with wt males, SP binds to sperm with which it enters the female. We wondered if sperm stored by mates of *SP*-null males, that lack bound SP, could become decorated with SP from a second male even if he did not provide sperm. If so, this would mean that SP from a second (spermless) male can bind to sperm from a prior male, already present in the female's reproductive tract (*Figure 1*. Cartoon).

To test whether SP from a second male can bind to SP-deficient sperm stored by mates of *SP*-null males, we first confirmed that no SP was detectable on sperm stored in females that had singly-mated to *SP*-null males (*Figure 1A*). We then examined whether SP was detected on such sperm if the female subsequently remated to a spermless male (who provided SP). We observed SP bound to the stored sperm following such rematings at either 1d (*Figure 1B*) or 4d (*Figure 1C*) after the original SP-less mating. We confirmed these findings with western blotting. Sperm stored in seminal receptacles of females that had mated to *SP*-null males and subsequently remated to spermless males were dissected and probed for the presence of SP. Consistent with our immunostaining data, SP was detected in samples of *SP*-null male's sperm from females that had remated to spermless males at 1d or 4d after the start of first mating (ASFM; *Figure 1D*, lanes 7 and 8). Thus, SP from a second male can bind to SP-deficient sperm stored from a prior male.

To see if mating order was important, we carried out the reciprocal cross, that is, testing if SP deposited by a first male (spermless, in this scheme) could bind to sperm that were subsequently introduced by a second (*SP*-null) male (*Figure 2*. Cartoon). Spermless males transfer SP to the female tract after mating (*Kalb et al., 1993*), but we did not detect any SP in females mated to spermless males by 1d after the start of mating (ASM; *Figure 1D*. lane 4). We saw no SP signal in samples isolated from females that had mated to spermless males, and then subsequently to *SP*-null males at 1d ASFM (*Figure 1D*. lane 5). Our immunofluorescence data were consistent with our western blots: we saw no SP-sperm binding in females that mated first with a spermless male and a day later with *SP*-null male (*Figure 2B*). Therefore, if SP entered the female without sperm, it was unavailable to bind to sperm from a subsequent SP-deficient male.

We hypothesized that we did not see SP bound to sperm in this second (reciprocal) crossing scheme because by the time of the second mating SP from the spermless male was no longer

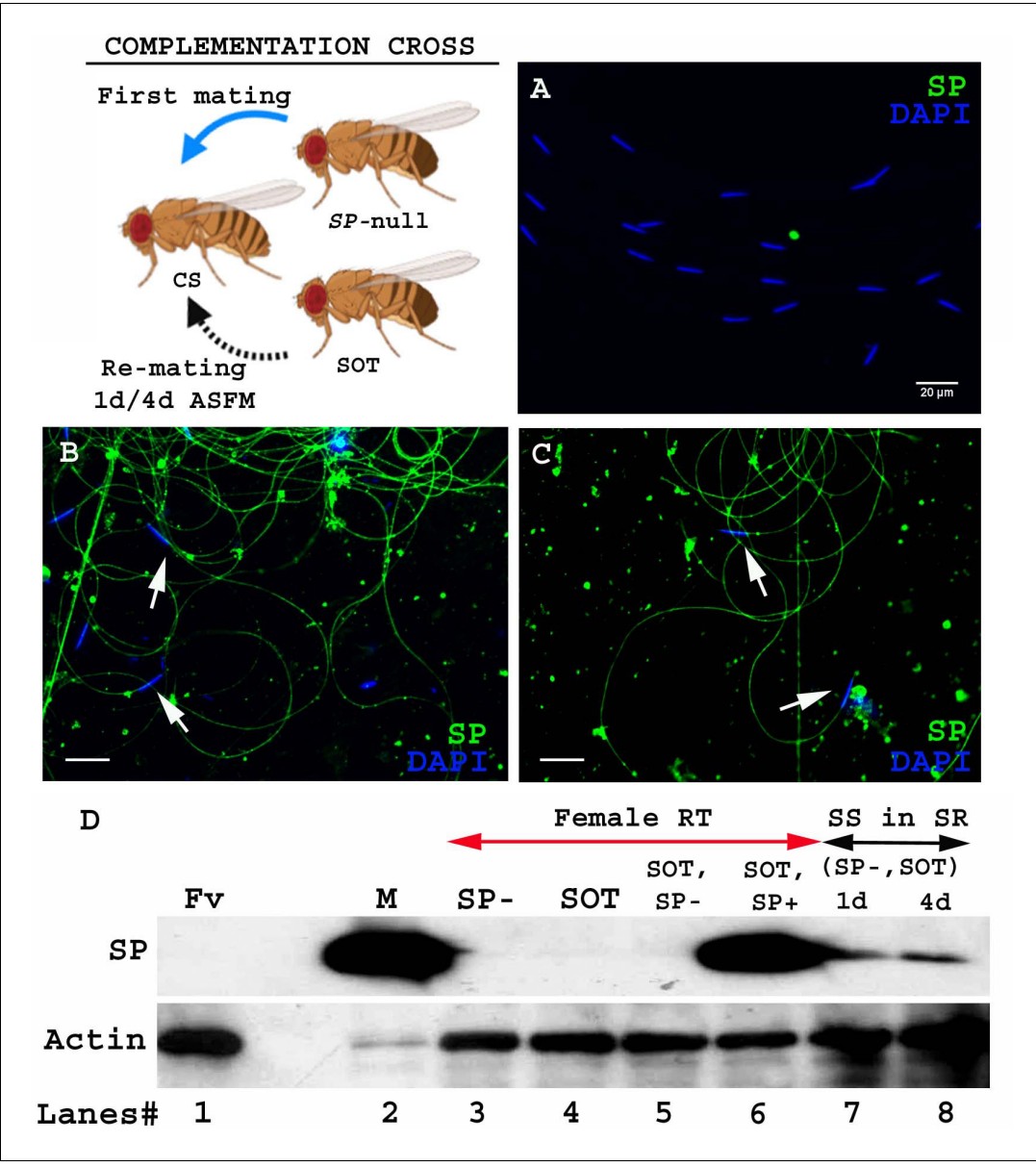

**Figure 1.** SP from a second male can bind to SP-deficient sperm of previous male stored within a mated female. Cartoon: Pictorial representation of the crossing scheme (fly images from Biorender). Wild type (CS) females were first mated to an *SP*-null male and then, at the indicated time, to a spermless (SOT) male. Sperm heads were stained with DAPI (blue) and SP was visualized with Alexa fluor 488, staining the sperm tail (green) and sperm head (cyan; overlapping blue/green). (A) Sperm from females singly mated to *SP*-null males, 1d ASM. (B) Sperm from females mated to *SP*-null males, remated to spermless males at 1d ASFM and (C) at 4d ASFM, both frozen 2 hr ASSM. White arrows indicate sperm heads. Bar = 20 µm (D) Western blot lane numbers 1: Fv, reproductive tract (RT) of virgin female (negative control; n = 5), 2: M, a pair of male accessory gland (positive control; n = 1), 3: SP-, reproductive tracts of females mated to *SP*-null males, 2 hr ASM (n = 5), 4: SOT, reproductive tracts of females mated to spermless males, 1d ASM (n = 5), 5: SOT, SP-, reproductive tract of females mated to spermless males and then remated to *SP*-null males, 1d ASFM (n = 8 RT), 6: SOT, SP+, reproductive tract of females mated to spermless males and then remated to control (SP+) males at 1d ASFM, frozen 2 hr ASSM (positive control; n = 8 RT), 7: (SP-, SOT), 1d and 8: (SP-, SOT), 4d sperm isolated from the seminal receptacle of females mated to *SP*-null males and then remated to spermless males at 1d ASFM and 4d ASFM, frozen 2 hr ASSM (n = 15 SS). Actin served as loading control.

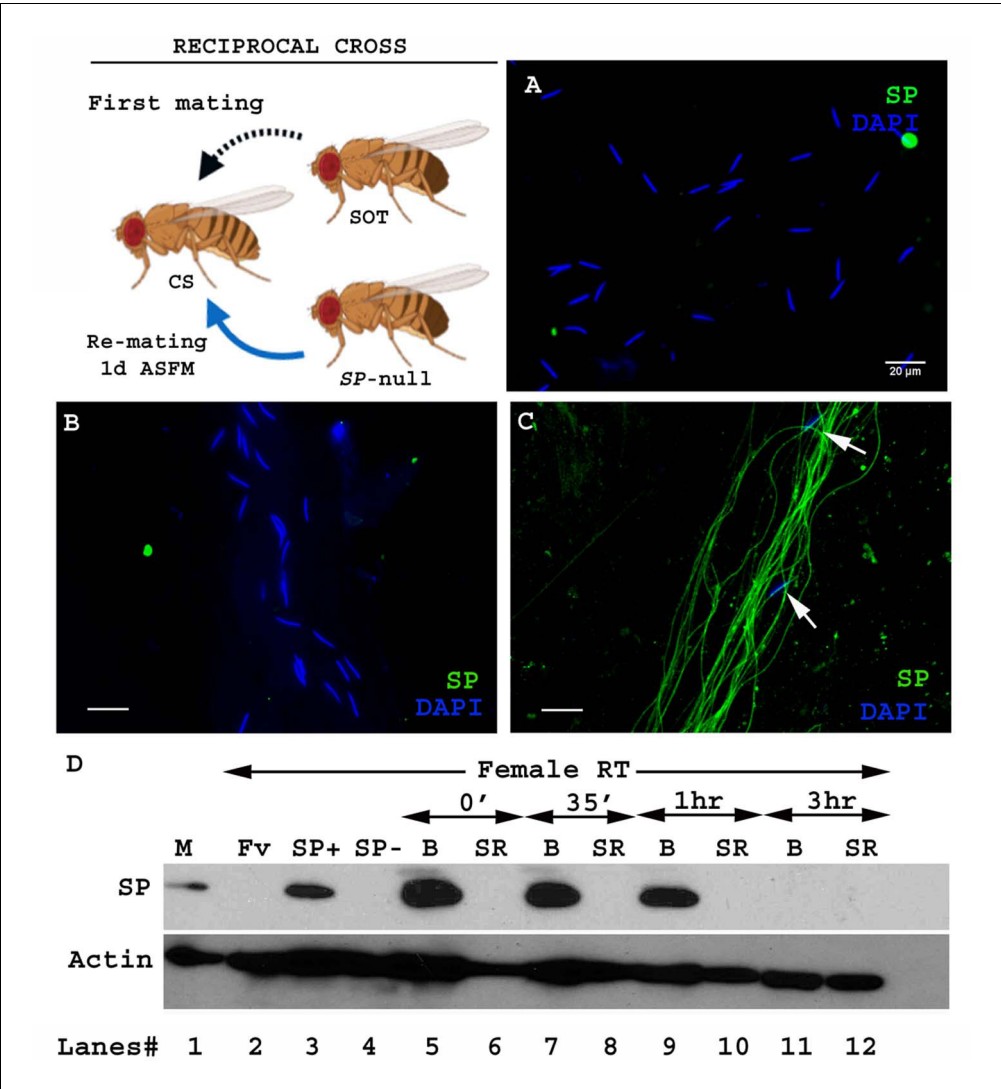

**Figure 2.** Sperm from a second male are not bound to SP from a prior spermless male. (Cartoon): Pictorial representation of the cross (fly images from Biorender); it is reciprocal of that in *Figure 1*. Females mated first with spermless (SOT) males and then a day later with *SP*-null males that provided sperm. Sperm heads were stained with DAPI (blue) and SP visualized with Alexa fluor 488, staining the sperm tail (green) and sperm head (cyan; overlapping blue/green). (**A**) Sperm from females singly mated to *SP*-null males, 2 hr ASM. (**B**) Sperm from females mated to spermless males and then remated to *SP*-null males, 1d ASFM. (**C**) Sperm from females mated to spermless males and then remated to SP+ males, 1d ASFM, serve as positive controls. Flies were frozen 2 hr ASSM. White arrows indicate sperm heads. Bar = 20 µm (**D**) Western blot lane numbers 1: M, a pair of male accessory gland (positive control; n = 1), 2: Fv, reproductive tract (RT) of virgin female (negative control; n = 5), 3: SP+, reproductive tract of females mated to control males (TM3 siblings of *SP*-null males; n = 5; positive control), 4: SP-, reproductive tract of females mated to *SP*-null males (n = 5; negative control). 5–12: Proteins from Bursa (B) or seminal receptacle (SR) from females mated to spermless males frozen at 0 min immediately after mating, 35 min, 1 hr, and 3 hr ASM, respectively (n = 15). Actin served as loading control.

The online version of this article includes the following figure supplement(s) for figure 2:

**Figure supplement 1.** Cartoon: Pictorial representation of cross (fly images from Biorender).

present in the female at 1d ASFM, since it could not be retained without binding to sperm (*Peng et al., 2005*) and no sperm were being supplied by these first males. To circumvent this, we attempted to remate females that had previously mated to spermless males as soon as 3–6 hr ASFM. However, few females remated, likely due to the recent experience of copulation, or to the effects of pheromones from the previous mating (*Shao et al., 2019*; *Laturney and Billeter, 2016*).

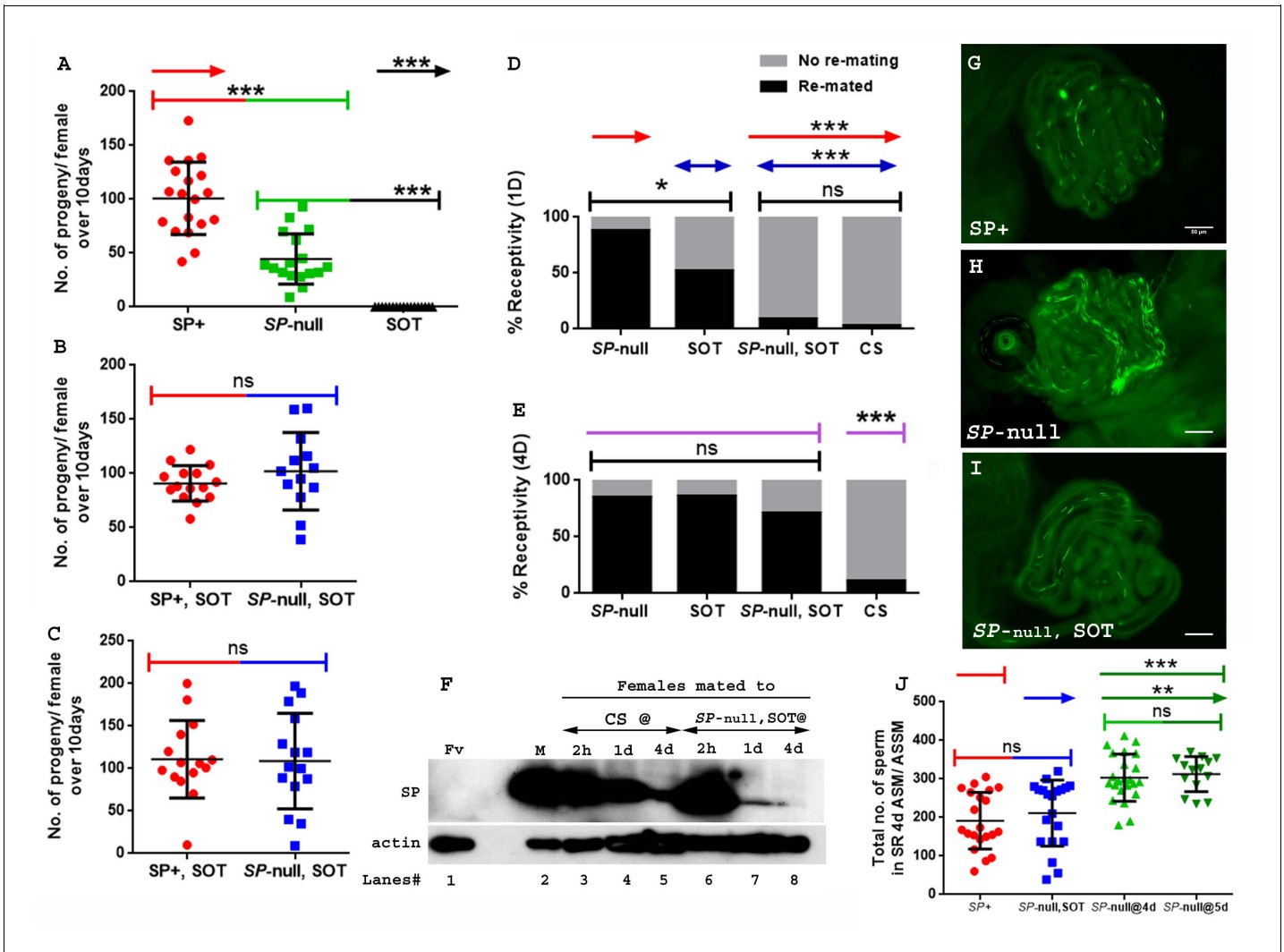

**Figure 3.** Remating with spermless males restores fertility, delays receptivity and optimizes efficient sperm release in females that previously mated to SP-null males. (**A**) Graphical representation of numbers of progeny produced by each female over the span of 10 days, following mating to control (TM3 siblings of SP-null males: SP+; red), SP-null males (SP-null; green), or spermless males (SOT), p***=<0.001; n = 15–20. (**B**) Fertility of females mated to SP-null males and then remated to spermless males at 1d ASFM (SP-null, SOT; blue, n = 15–20) and (**C**) Fertility of females mated to SP-null males and then remated to spermless males at 4d ASFM (SP-null, SOT; blue, n = 15–20) compared to females mated to control males and then remated to spermless males (SP+, SOT, red, ns = non significant). (**D**) Percentage receptivity of females mated to SP-null males and then remated to spermless males (SP-null, SOT) at 1d ASFM, when compared to females singly mated to SP-null males (red arrows), spermless (SOT, blue arrows) or CS males, 1d ASM (p*=<0.05; p***=<0.001; n = 15–20 for each technical replicate). (**E**) Percentage receptivity of females mated to SP-null males and then remated to spermless males (SP-null, SOT) at 4d ASFM, when compared to females singly mated to SP-null males, spermless (SOT) or CS males (purple arrows), 4d ASM (p***=<0.001; n = 15–20 for each technical replicate). (**F**) Western blot lane numbers 1: Fv, reproductive tract (RT) of five virgin females (negative control); 2: M, a pair of male accessory gland (positive control); 3, 4, 5: RT of females mated to CS males, flash frozen at 2 hr (n = 5), 1d (n = 15) and 4d (n = 15) ASM, respectively; 6, 7, 8: RT of females mated to SP-null males and then subsequently mated to spermless males at 1d ASFM, flash frozen 2 hr (n = 5), 1d (n = 15) and 4d (n = 15) ASSM, respectively. Actin served as loading control. (**G**) Sperm in the seminal receptacle (SR) of a typical female mated to a control male (SP+; ProtB-eGFP) at 4d ASM. (**H**) Sperm in the SR of a typical female mated to SP-null; ProtB-eGFP male at 4d ASM. (**I**) Sperm in the SR of a typical female, mated to SP-null; ProtB-eGFP and subsequently remated to a spermless male at 1d ASFM, and frozen at 4d ASSM. In (G–I) sperm heads are green due to eGFP. Bar = 50 µm. (**J**) Graphical representation of sperm counts in SRs of females singly-mated to control (SP+, red, TM3 siblings of SP-null; ProtB-eGFP), SP-null (green) or doubly-mated to SP-null and spermless male (SP-null, SOT, blue) represented in G, H, I panels (p**=<0.01; p*=<0.05; ns = non significant; n = 15–20).

The online version of this article includes the following source data and figure supplement(s) for figure 3:

**Source data 1.** Source data for *Figure 3D, E*.
**Figure supplement 1.** Western blot probed for SP.

In the few females that did remate, no SP-sperm binding was observed (*Figure 2—figure supplement 1*). Since the simplest explanation for these results was that SP transferred without sperm had disappeared from females by the time of the second mating, we performed western blotting to determine how long SP persists in the reproductive tract of females in absence of sperm. We probed for SP in proteins from the SR and bursa of females at 0 min, 35 min, 1 hr, and 3 hr ASM after mating. We detected SP in the bursa protein samples at 0 min, 35 min, and 1 hr ASM. (*Figure 2D*. lanes 5, 7, 9). However, SP was undetected in bursa or seminal receptacles of females at 3 hr ASM (*Figure 2D*. lane 11, 12). Thus, we could not determine whether SP from mating with a spermless male could bind a second male's sperm, because SP received from the first mating was lost from the female reproductive tract before a second mating could occur. *Xue and Noll, 2000* reported that a similar cross (females mated first to spermless males and then to *Prd* males) also gave no progeny (showed no copulation complementation), which they proposed to be due to inactivation or early loss of SFPs in the absence of sperm. Our results, showing that SP can bind to stored sperm from a prior male, provide the molecular explanation for their observation.

## SP from a second male restores fertility, inhibits receptivity and regulates optimal release of the first male's sperm from storage

SP is needed for efficient sperm release and utilization from the female sperm storage organs (*Avila et al., 2010*). We tested whether SP from a second male could restore the use of a first male's sperm. Females mated to spermless males have no progeny (*Figure 3A*). Females singly-mated to *SP*-null males have significantly reduced numbers of progeny (*Figure 3A*. SE of diff = 8.043; p\*\*\*=<0.001) relative to females mated to control males (*Figure 3A*), likely because lack of SP prevents the increase in egg production (*Chapman et al., 2003*; *Liu and Kubli, 2003*; *Chen et al., 1985*) and release of sperm from storage (*Avila et al., 2010*). However, females mated to *SP*-null males and then remated to spermless males at 1d (*Figure 3B*; p=0.2487) and 4d (*Figure 3C*; p=0.8618) ASFM had progeny levels similar to those of females that had mated to control (SP⁺) males and were subsequently remated to spermless males at the same time points. Thus, SP from a second (SOT) male could rescue the fertility defects that resulted from the lack of SP from an *SP*-null first male.

Reducing the likelihood of mated females to remate is another crucial postmating response regulated by SP (*Liu and Kubli, 2003*; *Chen et al., 1988*). Females that do not receive SP generally fail to exhibit this reluctance, and remate readily. We tested whether SP from a second male could delay the receptivity of females that had previously mated to *SP*-null males. Females singly-mated to *SP*-null males or spermless males show a significantly higher tendency to remate at 1d ASM (*Figure 3D*; p\*\*\*=<0.001) or 4d ASM (*Figure 3E*; p\*\*\*=<0.001) relative to females mated to wt (CS) males (*Figure 3D and E*). In contrast, females mated to *SP*-null males and then remated to spermless males at 1d ASFM (*Figure 3D*; p=0.43) showed receptivity similar to mates of control males at 1d after the start of second mating (ASSM). The effect, however, did not persist as long as after a mating to a wt male. At 4d ASSM (*Figure 3E*; p\*\*\*=<0.001) doubly-mated females exhibited higher receptivity relative to females mated to wt males but lower than those mated to spermless males. This could be either because less SP from the second (spermless) mating is able to bind to stored sperm from the previous mating and thus SP levels have been more depleted by 4 days ASSM than after a control mating where the sperm-SP enter the female together. Alternatively, the active portion of SP received from a rival male, bound to first male's sperm might be released from the sperm at a higher rate. We performed western blots to determine how long SP received from the second (spermless) male persists in the reproductive tract of females previously mated to *SP*-null males. Protein was extracted, and probed for SP, from females singly-mated to CS males and those doubly-mated to *SP*-null males and spermless males at 1d ASFM, or at 2 hr, 1d or 4d ASM/ASSM, respectively. SP signals were detected in females mated to CS males at 2 hr, 1d or 4d ASM (*Figure 3F*. lanes 3, 4, 5). SP was detected in females mated to *SP*-null males and then remated to spermless males at 2 hr and 1d ASSM (*Figure 3F*. lanes 6, 7) but not (or very weakly) at 4d ASSM (*Figure 3F*. lane 8). Taken together, our results show that SP from a second male can rescue the receptivity defects that resulted from the first male's of lack of SP but that sufficient SP for such an effect is not retained for as long as in a control situation (e.g. a mating with a wt male).

SP is also needed for release of sperm from storage within the mated female (*Avila et al., 2010*). Thus, females mated to *SP*-null males retain significantly more sperm in their seminal receptacle at

4d ASM. To test whether SP acquired from a spermless male in a second mating could also rescue this defect, we counted sperm in storage after a single mating with *SP-null; ProtB-eGFP* males and after mates of *SP-null; ProtB-eGFP* males had remated with spermless males. As expected, females mated to control (*SP+; ProtB-eGFP*) males had fewer sperm in their seminal receptacle (average of 192; *Figure 3G and J*) relative to mates of *SP-null; ProtB-eGFP* males, which had significantly higher sperm counts, indicating poor release of stored sperm (*Figure 3H and J*; p**=<0.01; average of 304 at 4d ASM). However, mates of *SP-null; ProtB-eGFP* males that had remated with spermless males retained sperm in numbers similar to those observed in females mated to control males (average of 212; *Figure 3I and J*; p=ns). We also counted sperm stored in seminal receptacle of females mated to *SP-null; ProtB-eGFP* males at 5d ASM (*Figure 3J*. average of 313) to make sure that the evident decline in sperm counts or release of stored sperm in doubly mated females (*SP-null; ProtB-eGFP* mates remated to spermless males and assayed at 5d ASFM or 4d ASSM) was not dependent on days after mating, but rather on receipt of SP from spermless males. Thus, SP from a second male can rescue the sperm release defects of prior mating to a male that lacked SP.

## SP from a second male can bind to stored sperm from a previous male, while still binding strongly to his own sperm

In the experiments described above SP was provided by a spermless second male, but in nature females are much more likely to encounter a male who has his own sperm, capable of binding his SP. To test whether SP from a male with sperm can still bind to sperm from another male, we modified our experimental protocol such that females were mated to *SP*-null males as described earlier, but rather than spermless males, we now used *ProtB-dsRed* males (*Manier et al., 2010*) as the second male (*Figure 4I*. Cartoon). These second males have a full suite of SFPs and sperm, and their sperm-heads are labeled with ProtB-dsRed. This allowed us to distinguish between sperm received from *SP*-null males (blue heads) and those received from *ProtB-dsRed* males (red heads). Females were frozen at 2 hr ASSM and sperm dissected from their seminal receptacles were probed for SP. We observed anti-SP staining along the entire sperm (head and tail) from *ProtB-dsRed* males (*Figure 4B*). Sperm received from the *SP*-null males (blue heads) were also stained with anti-SP along their length (head and tail; *Figure 4B*). Therefore, a control (wt) male with a complete suite of SFPs and sperm of his own can also provide SP to bind to SP-deficient sperm from another male.

The likelihood of finding an *SP*-null male in nature is very low. However, multiple-mating has been shown to deplete SFP reserves (*Hihara, 1981*), so it is possible that inter-ejaculate interaction could occur if the first male had depleted his SFP reserves. To test whether this could occur, we performed a crossing scheme in which we substituted multiply-mated control (CS) males with exhausted seminal reserves (*Hihara, 1981*) for the *SP*-null males used in *Figure 4A*. We carried out western blotting to determine the levels of SP in accessory glands (AG) of such multiply mated (CS) males and the amount of SP in their mates at 2 hr ASM. We observed relatively weak SP signals in the AG of multiply-mated males (*Figure 4—figure supplement 1*. A, lane 4) and a very faint SP signal in females mated to these males (*Figure 4—figure supplement 1*. A, lane 5) compared to relatively strong SP signal in virgin (unmated) males and the females mated to these males (*Figure 4—figure supplement 1*. A, lanes 2, 3 respectively). Our immunofluorescence data showed no (or extremely weak) SP-sperm binding in sperm dissected from the seminal receptacle of females mated to SFP-depleted males (*Figure 4—figure supplement 1*. C). Females mated to SFP-depleted CS males were then subsequently remated at 4d ASFM (long enough to have lost any SP signal from their first multiply-mated, mates) to *ProtB-dsRed* males. Sperm dissected from the seminal receptacles of these females at 2 hr ASSM were probed for SP (*Figure 4II*. Cartoon). There was no detectable SP signal on sperm stored in females singly-mated to SFP-depleted CS males at 4d ASM (*Figure 4C*). However, we observed anti-SP staining along the entire sperm (head and tail) received by the doubly-mated female from the SFP-depleted CS male (blue heads; *Figure 4D*) and *ProtB-dsRed* males (red + blue heads; *Figure 4D*).

Thus, in a normal mating, the amount of SP that a male transfers is sufficient to bind not only his own sperm but also to remaining sperm from a rival. Moreover, SP from an unmated control male can bind to previously stored sperm of a male that had his SFP reserves depleted prior to mating with the female.

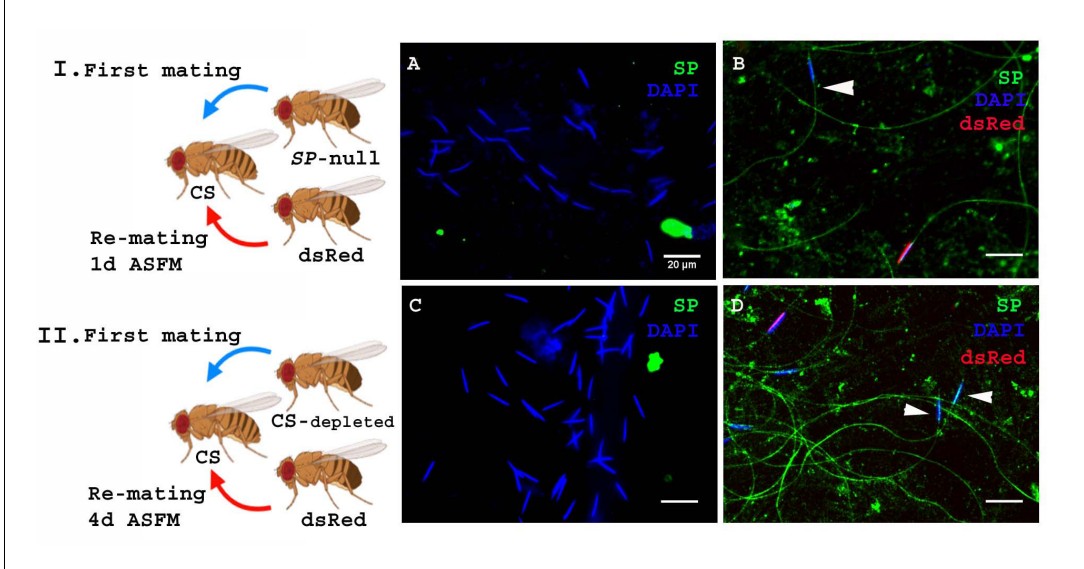

**Figure 4.** SP from a male who also provides sperm can bind to SP-deficient sperm as well as to the donor's sperm. Cartoon (I): Pictorial representation of the experimental cross (fly images from Biorender). Females mated to *SP*-null males were remated to control (*ProtB-dsRed*) males at 1d ASFM. (**A**) Sperm from females singly mated to *SP*-null males, 2 hr ASM (blue sperm-head). (**B**) Sperm from females mated to *SP*-null males (blue sperm-head) remated to *ProtB-dsRed* (red+ blue sperm-head) males at 1d ASFM. SP was visualized with Alexa fluor 488, staining the sperm (head+ tail) green. Flies were frozen 2 hr ASSM. White arrows indicate sperm heads (n = 10; Bar = 20 μm). Cartoon (II): Pictorial representation of the substitute cross (fly images from Biorender). Females mated to SFP depleted control (CS) males were remated to control (*Prot B-dsRed*) males at 4d ASFM. (**C**) Sperm from females singly-mated to SFP depleted CS males at 4d ASM (blue sperm-head). (**D**) Sperm from females mated to SFP depleted CS males (blue sperm-head), remated to *ProtB-dsRed* (red+ blue sperm-head) males at 4d ASFM. SP was visualized with Alexa fluor 488, staining the sperm (head+ tail)green. Flies were frozen 2 hr ASSM. White arrows indicate sperm heads (n = 10; Bar = 20 μm).

The online version of this article includes the following figure supplement(s) for figure 4:

**Figure supplement 1.** Little to no SP is transferred by multiply-mated males.

## Sex peptide binding to sperm of a prior male does not require receipt of LTR- SFPs from the second male

SP binding to sperm requires the action of a network of other SFPs- 'LTR-SFPs' (*Ravi Ram and Wolfner, 2009*; *Findlay et al., 2014*). Most of the known LTR-SFPs bind to sperm transiently (CG1656, CG1652, CG9997 and Antares) (*Singh et al., 2018*), while others do not bind to sperm (CG17575 or seminase; *LaFlamme et al., 2012*) the latter facilitate the localization of other LTR-SFPs, and SP, to the seminal receptacle. However, no LTR-SFPs are detectable on sperm or in female RT at 1d ASM (*Figure 5*). We wondered whether LTR-SFPs were required from the second male in order to bind his SP to the first male's sperm.

We carried out experiments similar to those previously described, in which females were first mated to *SP*-null males and then remated to spermless males at 1d ASFM. Sperm from the seminal receptacles of these females at 2 hr ASSM were immunostained for the presence of LTR-SFPs that had been received from second (spermless) males.

Females mated to CS males and frozen at 2 hr ASM served as positive controls for the sperm-binding of LTR-SFPsCG1656 (*Figure 5A*), CG1652 (*Figure 5E*) and CG9997 (*Figure 5I*). Females singly mated to *SP*-null males and frozen at 2 hr ASM exhibited normal sperm-binding of LTR-SFPs CG1656 (*Figure 5B*), CG1652 (*Figure 5F*) and CG9997 (*Figure 5J*), confirming that loss of SP affects neither the transfer nor the sperm-binding of other LTR-SFPs (*Singh et al., 2018*). By 1d ASM, stored sperm from females singly-mated to *SP*-null males showed no signal for the LTR-SFPs, CG1656 (*Figure 5C*), CG1652 (*Figure 5G*) and CG9997 (*Figure 5K*), as expected given the transient sperm-binding of these proteins (*Singh et al., 2018*). Thus, by the time these females remated with sperm-less males (1d ASFM), all known LTR-SFPs received from the first (*SP*-null) male were undetectable on sperm.

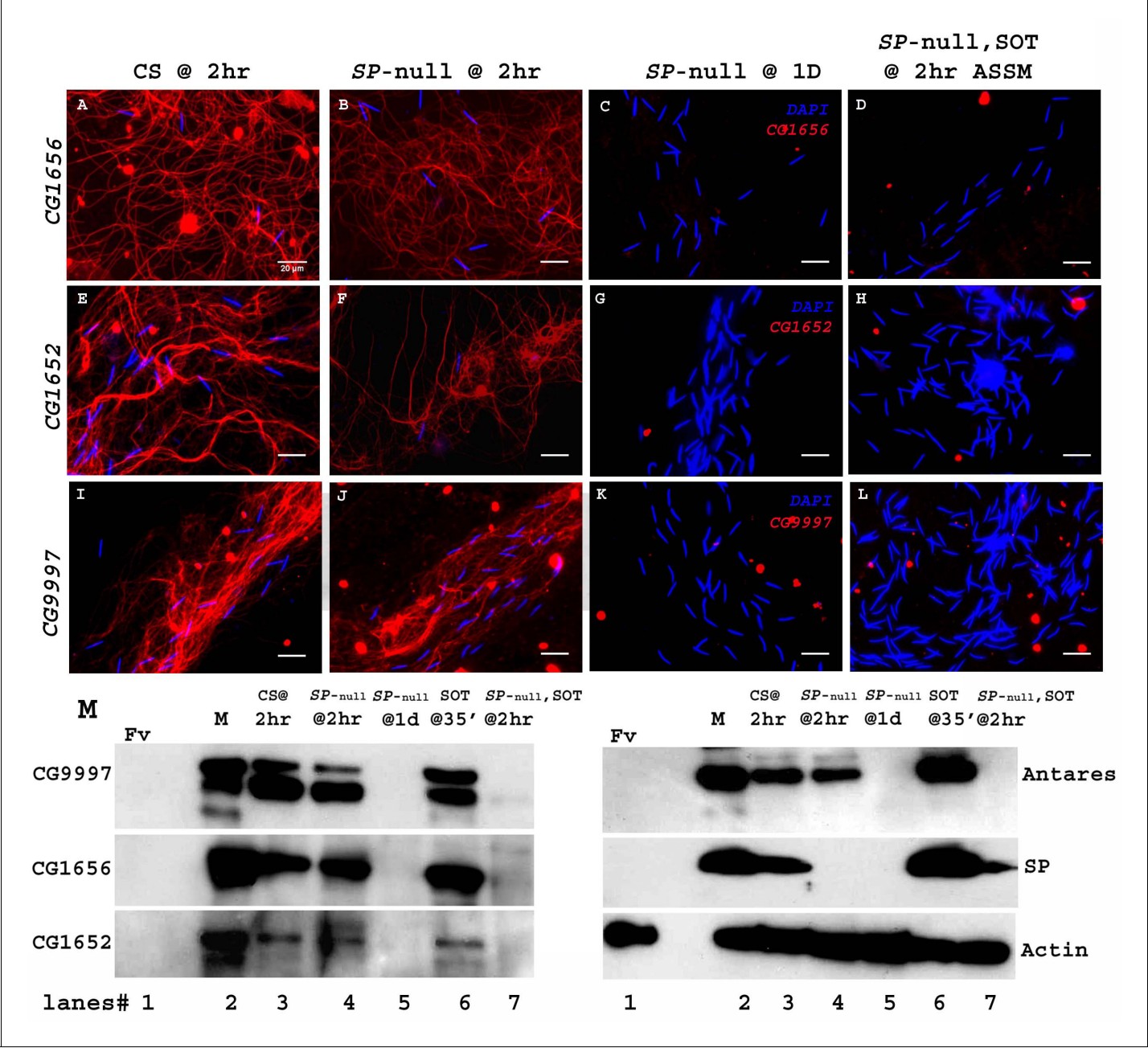

**Figure 5.** Sperm do not bind detectable LTR-SFPs from a second male. Females mated to wild type (CS) males at 2 hr ASM show LTR-SFPs bound to sperm, CG1656 (**A**), CG1652 (**E**), CG9997 (**I**). Females mated to *SP*-null males show the same (**B,F,J**) but by 1d postmating LTR-SFPs' signal were no longer detected on sperm (**C,G,K**) confirming previous reports (*Singh et al., 2018*). Females mated to *SP*-null males and then remated to spermless males also do not show detectable signal for sperm-LTR-SFP binding for CG1656 (**D**), CG1652 (**H**) and CG9997 (**L**), 2 hr ASSM, although they have SP bound (*Figure 1*). Sperm stained for the indicated LTR-SFP detected with Alexa fluor 594 (red) and sperm-head stained with DAPI (blue). Bar = 20 µm (**M**) Western blot probed for indicated LTR-SFPs. Lanes/samples are 1: Fv, reproductive tract (RT) of three virgin females (negative control); 2: M, one pair of male accessory glands (positive control); 3: CS @ 2 hr, sperm dissected from SR of 20 females mated to wild type (CS) males at 2 hr ASM; 4: *SP*-null @ 2 hr, sperm dissected from SR of 20 females mated to *SP*-null males at 2 hr ASM; 5: *SP*-null @1d, sperm dissected from SR of 20 females mated to *SP*-null males at 1d ASM; 6: SOT@35', reproductive tract of three females mated to spermless males at 35'ASM (positive control); 7: *SP*-null, SOT @ 2 hr, sperm dissected from SR of 20 females mated to *SP*-null males and then remated to spermless males at 1d ASFM, and frozen at 2 hr ASSM. Lanes were probed for LTR-SFPs CG9997, CG1656, antares and CG1652 and SP as described in the text. Actin served as loading control.

Interestingly, although females that mated to *SP*-null males and then to spermless males showed SP signal on their sperm (as in *Figure 1*) at 2 hr ASSM, we detected no signal of LTR-SFPs, CG1656 (*Figure 5D*), CG1652 (*Figure 5H*) and CG9997 (*Figure 5L*) on those sperm at 2 hr ASSM. This could be because LTR-SFPs from the second male could not enter the sperm storage organs in the absence of sperm or, alternatively, that their binding sites on sperm had been modified prior to the second mating (perhaps by the action of LTR-SFPs received from the first mating) to make them incapable of binding.

We verified these observations with western blots. Consistent with the immunofluorescence data in *Figure 5A–L*, LTR-SFP signals for CG1656, CG9997, CG1656 and Antares were detected in sperm dissected from females mated to CS and *SP*-null males at 2 hr ASM (*Figure 5M*. lanes 3, 4). No LTR-SFP signals were detected in sperm dissected from females mated to *SP*-null males at 1d ASM (*Figure 5M*. lane 5) or in sperm dissected from females mated to *SP*-null males, remated to sperm-less males at 1d ASFM, and frozen 2 hr ASSM (*Figure 5M*. lane 7). However, as expected SP signals were detected in sperm dissected from females that mated to *SP*-null males, remated to spermless males at 1d ASFM and frozen 2 hr ASSM (*Figure 5M*. blot probed for SP, lane 7).

Thus, sperm no longer detectably bind new LTR-SFPs after they have bound LTR-SFPs from their own (*SP*-null) male. That LTR-SFPs are needed for SP-sperm binding, and that SP from spermless male binds the first male's sperm, further suggests that the first male's sperm (or the female RT) had already been primed with its own LTR-SFPs during storage in the female tract.

Unlike the four LTR-SFPs assessed above, the two other LTR-SFPs, CG17575 and seminase, do not bind to sperm, yet are crucial for SFP-sperm binding. In the absence of CG17575 or seminase, SP fails to bind to sperm (*Ravi Ram and Wolfner, 2009*; *LaFlamme et al., 2012*). To determine if these proteins were required for a second male's SP binding to a first male's sperm, we first crossed females to *SP*-null males and then to *CG17575*-null or *seminase*-null males at 1d ASFM (*Figure 6*. Cartoon). In this situation, CG17575 and seminase had entered the female with the first male's sperm, but by the time of the second mating, were undetectable in the female (*Figure 6—figure supplement 1*). We examined whether in this situation SP transferred by *CG17575*-null (or *seminase*-null) males would still bind to the *SP*-null sperm stored in the female. We made use of ProtB-eGFP labeled *SP*-null males to differentiate between sperm received from first (cyan (DAPI+ eGFP) sperm heads) and second (blue (DAPI) sperm heads) males. Immunostaining and western blots for detection of SP on sperm dissected from females mated to *SP-null*; *ProtB-eGFP* males and then remated to *seminase*-null (*Figure 6*. A and C, lane 4) or *CG17575*-null (*Figure 6*. B and C, lane 5) males showed that SP received from the second male bound to sperm (head and tail) received from *SP-null*; *ProtB-eGFP* males. Sperm dissected from females singly-mated to *SP-null*; *ProtB-eGFP* males gave no SP signal, as expected (*Figure 6*. D, lane 3 and E) and sperm dissected from females singly-mated to *seminase*-null (*Figure 6*. D, lane 4 and F) or *CG17575*-null (*Figure 6*. D, lane 5 and G) males also showed no SP-sperm binding, as expected, due to lack of the LTR-SFP.

Therefore, sperm no longer require even CG17575 or seminase from the second male's ejaculate, after they have received the LTR-SFPs from their own (*SP*-null) male.

## Discussion

Ejaculate molecules, particularly the SFPs that are received by females during mating, play crucial roles in successful reproduction. In *Drosophila*, they induce striking changes in the physiology and behavior of females, instigating a wide array of post mating responses (*Avila et al., 2010*; *Scheunemann et al., 2019*; *Rubinstein and Wolfner, 2013*; *Avila and Wolfner, 2009*; *LaFlamme et al., 2012*; *Sitnik et al., 2016*; *Ravi Ram and Wolfner, 2007a*). Some of these responses persist long-term, due to binding of a male's SP to his sperm and gradual release of the SP's active C-terminal region (*Peng et al., 2005*). This important process is mediated by a cascade of 'LTR-SFPs' that are needed to bind SP to sperm (*Ravi Ram and Wolfner, 2009*; *Singh et al., 2018*; *Findlay et al., 2014*; *LaFlamme et al., 2012*). While all of the above can be seen as facilitating reproductive success of the mating pair (particularly from the male's perspective), SFPs also play roles in conflicts between males in species where females are polyandrous. *den Boer et al., 2010* investigated sperm survival in monoandrous and polyandrous ants and bees. They observed that while seminal fluid enhanced the survival of 'self' sperm, it preferentially killed the sperm of rival males. In other words, while SFPs worked in a cooperative interdependent way with 'self' sperm,

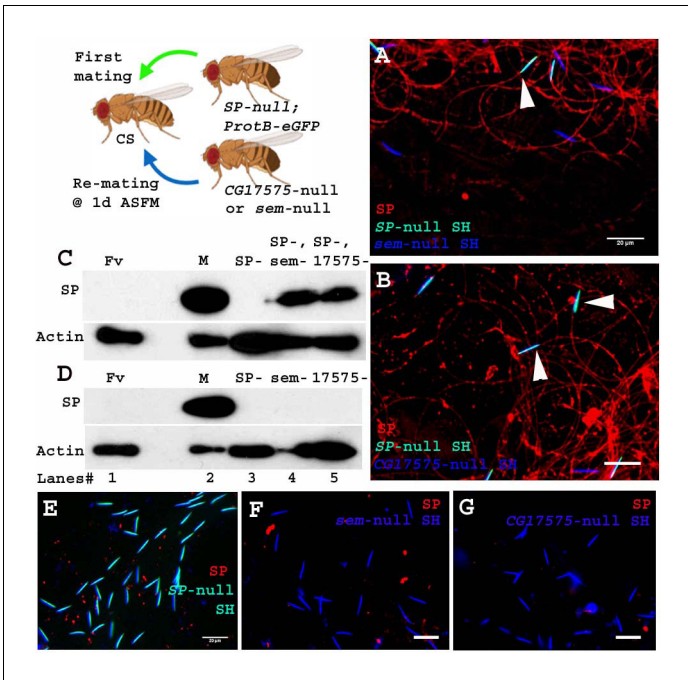

**Figure 6.** Sperm received from SP-null males do not require CG17575 or seminase from a second male to bind SP from that male. Cartoon: Pictorial representation of the experimental cross (fly images from Biorender). Females mated first with *SP-null; ProtB-eGFP* males [cyan sperm-head; DAPI(blue)+eGFP(green)] and then a day later with *CG17575*-null or *seminase*-null males (blue sperm-head; DAPI stained) and frozen, 2 hr ASSM. SP was visualized with Alexa fluor 594, staining the sperm (head+ tail) red. (**A**) Sperm from females mated to *SP-null; ProtB-eGFP* males and then remated to *seminase*-null males, 1d ASFM. (**B**) Sperm from females mated to *SP-null; ProtB-eGFP* males and then remated to *CG17575*-null males, 1d ASFM. (**C**) Western blot probed for SP. Lanes/samples are 1: Fv, reproductive tract (RT) of three virgin females (negative control); 2: M, one pair of male accessory glands (positive control); 3: SP-, sperm dissected from 20 females mated to *SP-null; ProtB-eGFP* males at 2 hr ASM; 4: SP-, sem-, sperm dissected from 20 females mated to *SP-null; ProtB-eGFP* males and subsequently to *seminase*-null males at 1d ASFM, frozen at 2 hr ASSM; 5: SP-, 17575-, sperm dissected from 20 females mated to *SP-null; ProtB-eGFP* males and subsequently to *CG17575*-null males at 1d ASFM, frozen at 2 hr ASSM. (**D**) Western blot probed for SP. Lanes/samples are 1: Fv, reproductive tract (RT) of three virgin females (negative control); 2: M, one pair of male accessory glands (positive control); 3: SP-, sperm dissected from 20 females mated to *SP-null; ProtB-eGFP* males at 2 hr ASM; 4: sem-, sperm dissected from 20 females mated to *seminase*-null males at 2 hr ASM 5: 17575-, sperm dissected from 20 females mated to *CG17575*-null males at 2 hr ASM. Actin served as loading control. (**E**) Sperm isolated from females singly mated to *SP-null; ProtB-eGFP* males, 2 hr ASM. (**F**) Sperm isolated from females singly mated to *seminase*-null male, 2 hr ASM. (**G**) Sperm isolated from females singly mated to *CG17575*-null male, 2 hr ASM. White arrows indicate sperm heads (represented as SH, n = 10; Bar = 20 μm).

The online version of this article includes the following figure supplement(s) for figure 6:

**Figure supplement 1.** Western blot probed for seminase and CG17575.

---

they harmed rival sperm when in a situation of conflict and cryptic female choice. Previous studies have shown that males respond to threat of rivals by altering their ejaculates, in terms of both sperm and non-sperm components (e.g. for *Drosophila*: [*Wigby et al., 2009*; *Garbaczewska et al., 2013*; *Sirot et al., 2011*]).

Studies of SFP functions have tended to investigate how a male's SFPs can promote the interests of his own (self) sperm. However, some data suggest that one male's SFPs (ovulin, ACP36DE) can indirectly benefit a subsequent male within a polyandrous female (*Neubaum and Wolfner, 1999*; *Rubinstein and Wolfner, 2013*; *Avila and Wolfner, 2009*; *Chapman et al., 2000*; *Nguyen and Moehring, 2018*). Here, we tested for direct effects of one male's SFPs on another male's sperm and/or fertility. Specifically, we show that SP from a second male can bind to and act with sperm received from a previous mating. Sperm stored in females mated to *SP*-null males show no SP-sperm binding (as expected), but if these mated females subsequently remate to a spermless male, his SP

can bind to stored sperm from the prior male. This binding of SP to the *SP*-null sperm restores his fertility and proper sperm release dynamics. Even if a second male transfers sperm, he transfers sufficient SP to bind to his own and rival sperm. Finally, our data suggest that the LTR-SFPs (that usually assist in binding of SP to sperm) are not required from the second male for the association of his SP with sperm received from the first male (who had already provided LTR-SFPs). The first male's sperm appear to be sufficiently 'primed' by prior receipt of their own LTR-SFPs to be able to bind SP from a second male.

## SP from a second male can associate with a prior male's sperm that were stored within the female

*Xue and Noll, 2000* reported that sperm transferred to females by *Prd* mutant males that lack the entire suite of SFPs were capable of fertilizing a few eggs to yield progeny, but only after the females were subsequently remated to spermless males. They coined the term 'copulation complementation' to describe this phenomenon, and proposed that SFPs from the second male might interact with the first male's sperm to yield this result. Consistent with this idea, several reports suggested that first-males that provided sperm but lacked particular SFPs (*Avila et al., 2010*; *Ravi Ram and Wolfner, 2007b*; *Mueller et al., 2008*; *Wong et al., 2008*; *Hopkins et al., 2019*) can have higher paternity shares in competitive situations: they were better competitors, compared to control males, in defensive sperm competition assays (*Avila et al., 2010*; *Avila and Wolfner, 2009*; *Fricke et al., 2009*). These reports align and are consistent with our observations that SP from a second male can bind to and assist the sperm from a prior SP-deficient male. A simple explanation for these results, based on the findings that we report here, is that the deficiency of these SFPs in the first male led to impaired release/use, and thus retention, of his sperm, and this was rescued by receipt of the second male's SFPs, as we have shown here, for SP.

SP is the only SFP thus far known to persist within the *Drosophila* female (for 10–14 days postmating), eliciting long-term post mating responses through gradual release of its C-terminal portion (*Peng et al., 2005*). The long-term persistence of SP on sperm made it an excellent candidate to examine for interaction with rival sperm. Here, we report that SP subsequently received from a spermless male binds to a first male's sperm (*SP*-null). This association is apparent even if the second mating occurs at 1d or as long as 4d ASFM, indicating that binding of SP to the first male's sperm occurs irrespective of how long sperm have been in the storage organs. It remains unclear how SP received from spermless (second) male enters the sperm storage organs, where sperm from the first mating had been stored. However, *Manier et al., 2010* reported that 60–90 min after the start of a second mating, 26% of the resident sperm (received from the previous mating) are moved from storage back into the bursa where they mix with the second male's ejaculate before moving back into the storage. Therefore, it is possible that SP received from the spermless male binds to the first male's sperm that relocated to the bursa, and the newly SP-bound sperm are then transferred back into storage in the seminal receptacle.

## The binding of SP received from one male to sperm of another can restore defects that resulted from lack of SP from the first male

In the absence of sperm, or if SP is not bound to sperm, females do not maintain post-mating responses and fail to efficiently release sperm from storage resulting in fewer sperm available for fertilization and fewer progeny (*Avila et al., 2010*; *Ravi Ram and Wolfner, 2009*; *Chapman et al., 2003*; *Liu and Kubli, 2003*). We observed that these defects were rescued when SP was received by females in a remating with spermless males. Thus, the second male's SP bound to the first male's sperm is functional. The rescue of the phenotype, however, was not as long lasting as in a normal single mating with SP transfer, wearing off by 4d postmating rather than the normal ~10 d. This could be because only fewer sperm relocated from storage to the bursa (*Manier et al., 2010*), so they may not carry sufficient SP back into storage to associate with *SP*-null sperm. Consistent with this, the levels of SP that we see stained in these situations are lower than those in a wild type mating.

## An unmated male transfers sufficient SP to bind to his own as well rival sperm

We did not know whether the amount of SP that is transferred during mating is more than the available binding sites on sperm. Here, we observed that an unmated control male does transfer enough SP to bind his own as well as pre-stored sperm (*SP*-null) in a previously mated female. Consistent with our findings, several reports suggest that in response to potential threats of sperm competition and conflicts, males adjust the levels of SFPs and transfer high amounts of SP, yet less ovulin, to previously mated females (*Wigby et al., 2009*; *Sirot et al., 2011*). *Rubinstein and Wolfner, 2013* demonstrated that ovulin induces ovulation, acting through octopamine (OA) neuronal signaling and increases the number of synapses that the female's Tdc2 neurons make on the musculature of the oviduct. Persistence of this latter effect could benefit rivals too, so second-mating males may thus be able to mitigate the levels of ovulin in their ejaculate. But the question remains that if SP from one male's ejaculate can bind to and assist another's sperm, why do males not lower the amount of SP transferred while mating? A potential explanation is that a male would still benefit by transferring enough SP to ensure that his own sperm remains saturated with SP, even at a cost of part of his SP binding to another male's sperm.

SP binds to sperm through its N-terminal region, and this region remains bound to sperm long-term (*Peng et al., 2005*). The bound N-terminal region of SP on sperm stored in a mated female does not allow any further binding of SP coming from rival male's ejaculate. Therefore under what circumstances might SP-mediated copulation complementation occur in nature? In polygynous males, SFPs are depleted faster than sperm (*Hihara, 1981*). This could result in a situation in which a female who mated with a male with low levels of SFPs might not receive enough SP to saturate his sperm. In these circumstances, SP received from another male would help compensate for the lower amount of SP from the depleted first male's ejaculate. SFP depletion would, of course, not only affect the levels of SP, but also all the other crucial LTR-SFPs. However, while other LTR-SFPs enable SP to bind sperm, it is the quantity of bound SP that correlates with the duration of post-mating responses. In line with this hypothesis, we subjected control males to recurrent matings (providing six virgins over the span of 2 days), with an intent to exhaust their SFPs. We observed that sperm stored by subsequent (7th) females mated to these multiply-mated males had undetectable SP signals. However, when these females were remated to unmated control males, strong SP signals were detected on both the SFP-depleted sperm received from the previous mating and the newly received rival sperm.

Therefore, our results support the idea that in nature males who have multiply-mated might get some help from the SFPs of subsequent, less depleted, males. Interestingly, this inter-ejaculate interaction might also confer an advantage to the second male. More of the second male's SP will be retained in the female reproductive tract, for even longer, if it binds to previously-stored sperm in addition to his own sperm. This could allow the post-mating responses in polyandrous females to be maintained for longer than in singly-mated females.

## Association of a second male's SP to sperm received from a prior male does not require the receipt of LTR-SFPs from the second male

Binding of SP to sperm is facilitated by a network of LTR-SFPs (*Ravi Ram and Wolfner, 2009*). Two LTR-SFPs, CG17575 and seminase, do not themselves bind to sperm, whereas other LTR-SFPs bind sperm transiently (CG1652, CG1656, CG9997, antares). CG17575 and seminase localize the other LTR-SFPs, and SP, to sperm storage organs (*Ravi Ram and Wolfner, 2009*; *Singh et al., 2018*; *LaFlamme et al., 2012*; *Ravi Ram and Wolfner, 2007a*; *Ravi Ram et al., 2006*). We found that SP from a second male (spermless or control) can associate with sperm from the first male (*SP*-null) even if it enters the female in absence of its own LTR-SFPs. This suggested that *SP*-null sperm (or the mated female RT) had already received modifications ('priming') from its own LTR-SFPs that were required for SP binding. This further suggests that once primed, a sperm can bind SP from a rival's ejaculate without the need for additional LTR-SFPs, and can restore its own post-mating dynamics.

Thus, we find that a critical SFP from one male can associate and offer direct benefits to sperm from another male, restoring the SP function to the previously stored sperm. Our work shows that SP is a crucial candidate for copulation complementation in *Drosophila*, and that sperm in storage

(or the female RT) are primed for SP binding by the first male's LTR-SFPs. Therefore, despite potential competition between males, there could be subtle cooperation between males as well. In addition, the allocation of resources by, and effects on, rival males that mate to polyandrous females, should be viewed in light of not only sexual conflicts, but also both direct and indirect effects of SFPs.

Additionally, our findings raise some intriguing questions for further study. First, our experiments, like those of *Xue and Noll, 2000*, were no-choice situations. SP from the second male had the opportunity to bind sperm only from one prior (SP-null) male. It will be interesting in the future to determine whether SP from a later male shows any preference to bind sperm from a more-related male, relative to sperm from a less-related one. Second, if there is such preference, its molecular mechanism is currently unknown and would be an important topic for further elucidation; we do not know what mechanisms distinguish self- from non-self sperm – whether molecular, temporal, or both. Third, although our data show that a male can benefit from the SP of a subsequent male, whether this is a true cooperation, or rather an accident of there being sufficient SP from the second male to bind to SP-deficient sperm in the female is unclear. It is possible that it is advantageous for a male to transfer large amounts of SP so as to coat his own sperm efficiently, even if this has the unintended consequence of there being sufficient SP to also bind to (and benefit) an SP-deficient rival's sperm. Alternatively, if related males are mating in proximity to each other, there may have been selection for such SP binding from a rival, if sperm from a male who was depleted of SFPs by prior mating bound SP that was likely from his relative. Each of these will be an intriguing topic for future investigation.

# Materials and methods

## Key resources table

| Reagent type (species) or resource | Designation | Source or reference | Identifiers | Additional information |
|---|---|---|---|---|
| Genetic reagent (*D. melanogaster*) | Tudor | R. Boswell; similar stock now available from Bloomington *Drosophila* Stock Center | BDSC:1735; FBst0001735; RRID:BDSC_1735 | FlyBase Genotype: *tud1 bw1 sp1*/*CyO* |
| Genetic reagent (*D. melanogaster*) | Δ325/TM3; Sb ry (SP-knockout line) | Gift from Eric Kubli | | |
| Genetic reagent (*D. melanogaster*) | Δ130/TM3; Sb ry (deficiency line) | Gift from Eric Kubli | | |
| Genetic reagent (*D. melanogaster*) | ProtB-eGFP(X); TM3/TM6 | Gift from Scott Pitnick | | |
| Genetic reagent (*D. melanogaster*) | ProtB-DsRed | Gift from Scott Pitnick | | |
| Antibody | anti-SP (rabbit polyclonal) | Wolfner lab | | IF (1:200), WB (1:2000) |
| Antibody | anti-CG1656 (rabbit polyclonal) | Wolfner lab | | IF (1:100), WB (1:1000) |
| Antibody | anti-CG1652 (rabbit polyclonal) | Wolfner lab | | IF (1:50), WB (1:500) |
| Antibody | anti-CG9997 (rabbit polyclonal) | Wolfner lab | | IF (1:50), WB (1:1000) |
| Antibody | IgG (H+L) Goat anti-Rabbit, Alexa Fluor 488 (goat anti-rabbit polyclonal) | Invitrogen | Cat. # A11008 RRID:AB_143165 | IF (1:300) |
| Antibody | IgG (H+L) Goat anti-Rabbit, Alexa Fluor 594 (goat anti-rabbit polyclonal) | Invitrogen | Cat. # A11012 RRID:AB_2534079 | IF (1:300) |

*Continued on next page*

*Continued*

| Reagent type (species) or resource | Designation | Source or reference | Identifiers | Additional information |
|---|---|---|---|---|
| Antibody | anti-Antares (rabbit polyclonal) | Wolfner lab | | WB (1:500) |
| Antibody | anti-seminase (rabbit polyclonal) | Wolfner lab | | WB (1:1000) |
| Antibody | anti-CG17575 (rabbit polyclonal) | Wolfner lab | | WB (1:1000) |
| Antibody | Anti-actin (mouse monoclonal) | Millipore Corp | Cat# MAB1501 RRID:AB_2223041 | WB (1:3000) |
| Antibody | Peroxidase AffiniPure Goat Anti-Rabbit IgG (goat anti-rabbit polyclonal) | Jackson Research | Code#111-035-003 RRID:AB_2313567 | WB (1:2000) |
| Antibody | Peroxidase AffiniPure Goat Anti-Mouse IgG (goat anti-mouse polyclonal) | Jackson Research | Code#115-035-003 RRID:AB_10015289 | WB (1:2000) |
| Other | DAPI stain | Invitrogen | Cat. # PI62247 | (1 µg/mL) |
| Other | Poly-L-Lysine (0.1 % w/v in $H_2O$) | Sigma | P8920-100ML | 0.01% w/v in $H_2O$ |
| Other | Albumin from Bovine Serum (BSA) | Sigma | A9418-50G | 5% in 1X PBS |
| Other | CitiFluor Mountant Solution | Electron Microscopy Sciences | Cat. #17970–100 | |
| Software, Algorithm | Graph Pad Prism | | RRID:SCR_002798 | Version 6.01 |

## Fly strains

Spermless males, [*sons of tudor,* (SOT) that lack sperm but produce and transfer a complete suite of SFPs] were the progeny of *bw sp tud[1]* females (*Boswell and Mahowald, 1985*) mated to control, *Canton S* (CS) males. *Sex peptide* null mutant males ($\Delta325/\Delta130$; which have sperm and the entire suite of SFPs except for SP) (*Liu and Kubli, 2003*) were generated by crossing the SP knockout line ($\Delta325/TM3, Sb ry$) to a line carrying a deficiency for the *SP* gene ($\Delta130/TM3, Sb ry$). Control males were the TM3 siblings of *SP*-null mutants. Matings were conducted with wild type *D. melanogaster* females (CS). To determine sperm numbers, we generated a line carrying the *SP*-null mutation and Protamine B-eGFP tagged sperm (*ProtB-eGFP/Y; $\Delta325/\Delta130$*) by series of crosses between the *SP* knockout line ($\Delta325/TM3, Sb ry$) and *ProtB-eGFP (X); TM3/TM6* (*Manier et al., 2010*). The *TM3* siblings of these males, (*SP[+]; ProtB-eGFP*) served as controls. Sperm-heads of these control males were tagged with ProtB-eGFP, but the males had normal levels of SP (*Figure 3—figure supplement 1*). *ProtB-ds Red* males with Protamine B-dsRed tagged sperm heads (*Manier et al., 2010*) served as additional controls. All flies were reared under a 12:12 hr light-dark cycle at 22 ± 1°C on standard yeast-glucose medium. Mating experiments were carried out by single-pair mating 3–5 day old virgin CS females to 3- to 5-day-old unmated males of genotypes indicated in the text and remating the same female 1 day or 4 days after the start of first mating (ASFM) to age matched unmated males of the genotypes indicated in the text.

## Crossing scheme to study first male's sperm and rival's SP binding

*Xue and Noll, 2000* reported copulation complementation in females mated to *Prd* males (which produce sperm but lack SFPs) remated to spermless males that produce SFPs. We followed a similar scheme but to focus on SP specifically, we used *SP*-null males as the first male. As described in Results, we then remated these females to spermless males, which make SFPs but not sperm. We attempted to do the reciprocal experiment, where females were mated to spermless males and then remated to *SP*-null males, but consistent with what was reported by *Xue and Noll, 2000*, we could not detect copulation complementation in this direction for technical reasons: SP from the spermless

male did not persist long enough in the mated female to interact with the second male's sperm (see Results). We carried out rematings at three time points, 3–6 hr, 1d, and 4d AFSM. We assessed results at 2 hr after the start of the second mating (ASSM).

## Fertility

The reproductive performances of singly-mated or doubly-mated females were assayed by analyzing fertility (numbers of progeny eclosed over ten days) (*Kalb et al., 1993*). Briefly, the fertility assays were carried out with (A). 'Single matings': Females were singly mated to (i) spermless males, (ii) *SP*-null males, or their (iii) *TM3* siblings (genetically-matched control males) in three individual sub-batches, and (B). 'Rematings': Females were mated to *SP*-null males or their *TM3* siblings (*SP+*) and were then subsequently remated to spermless males at 1d and 4d ASFM. Matings that lasted 15 mins or more were considered successful. At the end of a mating, males were removed from the vials and females were allowed to lay eggs for 10 days after the start of mating (ASM) in the first batch and after the start of second mating (ASSM) in the second batch. Females were transferred to fresh food vials every 3 days. Flies emerging from each vial were counted. Fertility is represented as total number of progeny produced by each female over a period of 10 days. The differences in fertility were analyzed through one-way Analysis of Variance (ANOVA) followed by Tukey's multiple comparison tests for single-matings and Mann Whitney U-tests for rematings. All assays were repeated more than two times and comprised of two technical replicates, with each group consisting of a minimum sample size of 15–20.

## Receptivity

To determine the propensity of females to remate, receptivity assays (*Chapman et al., 2003*) were set for females singly mated to *SP*-null, spermless or CS males and females mated to *SP*-null males and then subsequently remated to spermless males at 1d ASFM. For the assay, females from singly-mated and doubly-mated groups were then provided with (CS) males at 1d and 4d ASM or ASSM, respectively. We determined the number of females that mated within 1 hr from when the CS male was introduced within the vial. Assays were repeated more than two times, with each group consisting of a minimum sample size of 15–20. The data were analyzed by Fisher exact tests and Chi-squared group analyses.

## Sperm utilization/release from sperm storage organs in females

To study the effect of first male's sperm and rival male's SP binding on sperm utilization and release, we generated *SP*-null males whose sperm-heads are labelled with ProtB-eGFP (*Manier et al., 2010*). Females were mated to *SP-null; ProtB-eGFP* or *SP+; ProtB-eGFP* (control) males. Some of the mated females were frozen at 4d ASM (or 5d ASM) for sperm counts. The remaining mates of *SP-null; ProtB-eGFP* males were remated to spermless males at 1d ASFM. These flies were frozen at 4d ASSM. Subsequently, seminal receptacles of females singly-mated to *SP-null; ProtB-eGFP* and *SP+; ProtB-eGFP*, or doubly-mated to *SP-null; ProtB-eGFP* and spermless males, were dissected and eGFP sperm were counted (at a total magnification of 200X, with FITC filter on an Echo-Revolve microscope). Mature sperm in the seminal receptacles of mated females were counted twice and groups were blindfolded to ensure reproducibility and avoid bias. The percent repeatability was 90–94%. Assays were repeated more than two times, with two technical replicates. Every group contained a minimum sample size of 15–25. Differences in the sperm counts between groups were analyzed statistically through one-way ANOVA followed by Tukey's multiple comparison tests.

## Brood matings

Control (CS) males were subjected to brood matings (*Misra et al., 2014*; *Gilchrist and Partridge, 1995*) to deplete SFPs, as their levels are known to become exhausted at a higher rate than sperm numbers (*Hihara, 1981*). Briefly, 3-day-old control males were mated to CS females in two broods (each consisting of three virgin females) over 2 days. The first mating of both broods was observed. On the third day, previously mated females were removed and the male was provided with an additional virgin female (7th mate), matings were observed and depleted CS males were removed. Half of the 7th mated females were frozen at 4d ASM, while the others were subsequently remated to

control (*ProtB-dsRed*) males at 4d ASFM, and then frozen at 2 hr ASSM. Sperm stored in the seminal receptacle of the frozen flies were dissected and immunostained for SP.

## Immunofluorescence

Immunofluorescence was performed to detect SP-sperm binding (*Ravi Ram and Wolfner, 2009*; *Peng et al., 2005*; *Singh et al., 2018*). Sperm dissected from seminal receptacles of experimental or control females were attached to poly-L-Lysine (Sigma) coated slides. Sample processing was carried out according to the protocol of *Ravi Ram and Wolfner, 2009* with minor modifications. Samples were blocked with 5% bovine serum albumin (BSA) in 1X PBS for 30 min. Subsequently, samples were incubated overnight in rabbit anti-SP(1:200), CG1656(1:100), CG1652(1:50), CG9997(1:50) (*Singh et al., 2018*), in 0.1% BSA at 4°C overnight. Samples were then washed in PBS and incubated at room temperature for 2 hr in goat anti-rabbit IgG coupled to alexa fluor 488 (green) or 594 (red; Invitrogen) at a concentration of 1:300 in 1x PBS at room temperature in the dark. Samples were then washed in PBS, incubated in 0.01% DAPI for 3 min at room temperature in the dark, rewashed and mounted using antifade (CitiFluor mountant solution; EMS). The fluorescence was visualized under an Echo-Revolve fluorescence microscope at a magnification of 200X. A minimum of three independent immunostaining batches, with a minimum sample size of 10, were analyzed for each group.

## Sample preparation and western blotting

To further examine transfer, persistence or binding of SP to sperm stored in singly-mated or doubly-mated females, the lower reproductive tract (RT) or sperm stored (SS) in seminal receptacles of mated female were dissected. The dissected tissues (lower RT, n = 5–10 or sperm, n = 20–30) were suspended in 5 µl of homogenization buffer (5% 1M Tris; pH 6.8, 2% 0.5M EDTA) and processed further according to the protocol of *Ravi Ram and Wolfner, 2009*. Proteins from stored sperm or lower female reproductive tract were then resolved on 12% polyacrylamide SDS gel and processed further for western blotting. Affinity purified rabbit antibodies against SP(1:2000), CG1656(1:1000), CG1652(1:500), antares(1:500), CG9997(1:1000), CG17575(1:1000), seminase(1:1000) (*Ravi Ram and Wolfner, 2009*; *Singh et al., 2018*; *LaFlamme et al., 2012*) and mouse antibody against actin (as a loading control; Millipore Corp., cat no. #MAB1501 at 1:3000) were used as primary antibodies. HRP conjugated secondary anti-rabbit and anti-mouse antibodies (Jackson Research) were used for detection of SFPs at a concentration of 1:2000.

## Acknowledgements

We thank Dr. Ravi Ram Kristipati, S Allen, N Brown, S Delbare, D Chen, Y Yamashita, and an anonymous reviewer for helpful suggestions and comments on the manuscript, and N Buehner for technical advice. We are grateful to the NIH for grant R01-HD038921 to MFW, which supported this work.

## Additional information

### Funding

| Funder | Grant reference number | Author |
|---|---|---|
| Eunice Kennedy Shriver National Institute of Child Health and Human Development | R01-HD038921 | Mariana F Wolfner |

The funders had no role in study design, data collection and interpretation, or the decision to submit the work for publication.

### Author contributions

Snigdha Misra, Conceptualization, Data curation, Formal analysis, Investigation, Visualization, Methodology, Writing - original draft, Writing - review and editing; Mariana F Wolfner, Conceptualization, Resources, Formal analysis, Funding acquisition, Writing - original draft, Project administration, Writing - review and editing

**Author ORCIDs**

Mariana F Wolfner (iD) https://orcid.org/0000-0003-2701-9505

**Decision letter and Author response**

Decision letter https://doi.org/10.7554/eLife.58322.sa1

Author response https://doi.org/10.7554/eLife.58322.sa2

## Additional files

**Supplementary files**

• Transparent reporting form

**Data availability**

All data generated or analysed during this study are included in the manuscript and supporting files. Source data files have been provided for Figure 3D,E.

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
