## [Decision Letter]

**Acceptance summary:**

This manuscript by Misra and Wolfner reports an interesting study on inter-ejaculate interaction in *Drosophila*, identifying underlying molecular mechanism of this phenomenon. They show that seminal fluid protein SP can bind to other males sperm, 'benefiting' other males. In the original submission, the reviewers raised the concerns regarding the balance of the Discussion, whether or not SP transfer is truly benefitting others, or there may be other explanations. In the revision, the authors addressed these issues by modifying their statements and Discussion. This paper will make an important contribution to the field.

**Decision letter after peer review:**

Thank you for submitting your article "*Drosophila* seminal Sex Peptide can bind rival as well as own sperm and provide function for SP in polyandrous females" for consideration by *eLife*. Your article has been reviewed by two peer reviewers, including Yukiko M Yamashita as the Reviewing Editor and Reviewer #1, and the evaluation has been overseen by K VijayRaghavan as the Senior Editor.

The reviewers have discussed the reviews with one another and the Reviewing Editor has drafted this decision to help you prepare a revised submission.

Summary:

This manuscript by Misra and Wolfner reports an interesting study on inter-ejaculate interaction in *Drosophila*, identifying underlying molecular mechanism of this phenomenon. The reviewers identified overlapping set of concerns, which can be addressed by textual editing. (if the authors can add any experiments that strengthen the paper, instead of tempering the statement, it would be welcome, but under current circumstances, it is not expected).

But reviewers felt that more balanced discussions may be warranted. In particular, the reviewers felt that "copulation complementation" suggests "cooperation" and the authors need to be extra careful here. Cooperation is almost certainly not an evolutionarily stable solution (very rare, requires some kind of enforcement or genetic relatedness, etc.). Maybe what we are observing is simply the limits of the system, that SP cannot help but bind other male's sperm, and males don't really have a solution for that. We suggest that the authors incorporate these (and related issues in individual comments) to provide careful and balanced discussion.

Reviewer #1:

Misra and Wolfner describe their study about *Drosophila* sex peptide (SP) in the context of inter-ejaculate interaction. It was known that certain sterile males can sire progeny, if their mates (female) mate other males in a phenomenon called copulation complementation, suggesting that 'something' is transferred from the second male to the first male's sperm to help fertilization by the first male's sperm.

In this study, the authors identify SP as the 'something' that is provided by the second male to potentiates the first male's sperm, identifying the molecular basis of copulation complementation. This also has an implication that, in nature, males might benefit their rivals, an interesting new parameter in the context of sperm competition and evolution.

They show that:

– SP from the second male (spermless) can associated with the sperm of the first male (which does not have SP), making the first male's sperm competent for fertilization (but SP is short-lived after being transmitted to female, so the opposite mating scheme did not complement).

– This is likely relevant in nature as well, because multiple-mating reduces SFPs in males, resembling SP-deficient males. And this idea was tested experimentally in later figures of the paper.

This is a well-conceived and well-executed study, and significantly contributes to the field.

Reviewer #2:

The authors demonstrate that sex peptide from one male can bind to sperm from another male, and that this is associated with some of the same aspects of fertility such as increased female egg laying and decreased female remating propensity. This is an excellent paper, extremely well written and clear. There are a number of very interesting observations, and *eLife* seems like an appropriate venue.

1) My most major set of comments centers around the fact that all of these are essentially "no-choice" experiments, and this has two main meanings.

a) First, SOT males do not make sperm so their SP has no "choice" but to bind to the other male's sperm. Therefore, although it is clear that SP from one male can bind to another male's sperm, a major question remains: does SP "prefer" to bind sperm from its own male if given a choice? Is it possible to mix sperm from two different males in vitro, add SP from one of the males, and quantify the amount of SP that binds to different sperm? Subsection “SP from a second male can bind to stored sperm from a previous male, while still binding strongly to his own sperm” of the Results, relaxes this "no-choice" aspect, by showing that SP from one male can still bind another male's sperm even when both males make sperm. However, there could still be some kind of preference for one sperm or the other.

b) Second, the females in this experiment had no "choice" of sperm. Although SP from one male can restore some fertility-associated phenotypes (i.e., their Figure 3D), an important question is whether sperm are just as functional if their SP comes from a different male. A sperm competition experiment seems important here:

Female crossed to male from genotype 1, then crossed to male from genotype 2 (both males make SP). Repeat, but with a male from genotype 1 that has SP-null allele crossed into it. Genotype offspring. This experiment would allow comparison of P1 when the first male can vs. cannot produce its own SP. It is quite possible that the sharing of SP shown here has no effect on real fitness under sperm competition, or even that the binding which is clearly shown in Figure 1 actually has some detrimental effect to the other male's sperm. The authors may have strains already in the lab that could be used in such an experiment.

A counter-argument to my entire comment #1 is that a major goal in the current paper is to understand the molecular basis of "copulation complementation" from the Xue and Noll paper. If this is the goal the paper succeeds 100%. But "choice" experiments would certainly increase interpretation of the evolutionary significance of these results. If "choice" experiments are not possible, I would at least recommend some discussion of how they might lead to additional insight.

2) If SP is a protein that binds sperm, is it that surprising that SP binds sperm from other males? At the outset, the null hypothesis seems to be that a protein that binds sperm will bind another male's sperm. What seems more interesting are the experiments that showed that the LTR-SFPS bind in sperm specific manner (subsection “Sex peptide binding to sperm of a prior male does not require receipt of LTR- SFPs from the second male” of Results, especially around the fourth paragraph). But this part of the paper is kind of tucked away, for example does not appear in the Abstract.

3) One could imagine that genetic relatedness among males is an important facet here. In the trivial case where the two male mates are genetically identical, SP would not be able to discriminate between self vs. non-self sperm. I realize that is not the experiment they did, and that in fact the two males in Figure 1 were derived from distinct strains. But my question is, how genetically distinct are they? Some more discussion of the genetic backgrounds is warranted. And what might happen if they used males that was more versus less genetically related? Might they see a difference in the sharing of SP? The reason I ask this question is because if this is some kind of "cooperation", it should be more apparent among related individuals. This might not be addressable, but might be useful to add to the Discussion.

---

## [Author Response]

Summary:This manuscript by Misra and Wolfner reports an interesting study on inter-ejaculate interaction in *Drosophila,* identifying underlying molecular mechanism of this phenomenon. The reviewers identified overlapping set of concerns, which can be addressed by textual editing. (if the authors can add any experiments that strengthen the paper, instead of tempering the statement, it would be welcome, but under current circumstances, it is not expected).But reviewers felt that more balanced discussions may be warranted. In particular, the reviewers felt that "copulation complementation" suggests "cooperation" and the authors need to be extra careful here. Cooperation is almost certainly not an evolutionarily stable solution (very rare, requires some kind of enforcement or genetic relatedness, etc.). Maybe what we are observing is simply the limits of the system, that SP cannot help but bind other male's sperm, and males don't really have a solution for that. We suggests that the authors incorporate these (and related issues in individual comments) to provide careful and balanced discussion.

We are very grateful for the insightful comments of the reviewers and Reviewing Editor. They rightfully point out ways in which our discussion needed more balance. We have revised the paper accordingly, including taking out many of the mentions of “copulation complementation” (we left a couple in, since it was not our term and we are trying to connect to it). We have added the balancing-comments to (especially) the Discussion. We appreciate them, as we agree it adds more balance.

Reviewer #2:The authors demonstrate that sex peptide from one male can bind to sperm from another male, and that this is associated with some of the same aspects of fertility such as increased female egg laying and decreased female remating propensity. This is an excellent paper, extremely well written and clear. There are a number of very interesting observations, and eLife seems like an appropriate venue.1) My most major set of comments centers around the fact that all of these are essentially "no-choice" experiments, and this has two main meanings.a) First, SOT males do not make sperm so their SP has no "choice" but to bind to the other male's sperm. Therefore, although it is clear that SP from one male can bind to another male's sperm, a major question remains: does SP "prefer" to bind sperm from its own male if given a choice?

This is an excellent point. To our knowledge no studies have addressed this to date, and our current lab-situation would make it difficult at best to do so. Instead, we modified the text to acknowledge this as an important possibility to consider, and an area for future study.

Is it possible to mix sperm from two different males in vitro, add SP from one of the males, and quantify the amount of SP that binds to different sperm?

This is a very interesting idea, but unfortunately we cannot do the experiment for a technical reason. Our recent data show almost all binding of SP to sperm only occurs once both are within the reproductive tract of the female. We don’t yet know what is required from the female tract environment, not do we know the ionic or pH conditions in the female reproductive tract. This makes us concerned that in vitro experiments may not be able to replicate the true milieu well enough to be certain of the interpretation of the results. Getting a robust in vitro system is among our current priorities, but we are not at a position to be able to carry out this experiment.

Subsection “SP from a second male can bind to stored sperm from a previous male, while still binding strongly to his own sperm” of the Results, relaxes this "no-choice" aspect, by showing that SP from one male can still bind another male's sperm even when both males make sperm. However, there could still be some kind of preference for one sperm or the other.

Agreed. We now mention this in the Discussion, as an interesting possibility that should be tested in the future. Thank you.

b) Second, the females in this experiment had no "choice" of sperm. Although SP from one male can restore some fertility-associated phenotypes (i.e., their Figure 3D), an important question is whether sperm are just as functional if their SP comes from a different male. A sperm competition experiment seems important here:Female crossed to male from genotype 1, then crossed to male from genotype 2 (both males make SP). Repeat, but with a male from genotype 1 that has SP-null allele crossed into it. Genotype offspring. This experiment would allow comparison of P1 when the first male can vs. cannot produce its own SP. It is quite possible that the sharing of SP shown here has no effect on real fitness under sperm competition, or even that the binding which is clearly shown in Figure 1 actually has some detrimental effect to the other male's sperm. The authors may have strains already in the lab that could be used in such an experiment.

Avila et al. conducted this experiment, and found that when competed against a common second-male, SP null males were had a higher P1 than SP+ controls in the same situation. We believe that this is because mates of SP null males retain more first-male sperm than normal (because of the lack of SP), giving the SP null males an advantage over SP+ controls when these males are the first males to mate. The SP null males are less fit overall (having fewer progeny than normal, etc.), but in this particular situation they do “better” than controls because of the abnormal retention of their sperm; we did not mean to imply that the rescue make them more fit. We have added text to the Introduction and the Discussion, to highlight this.

A counter-argument to my entire comment #1 is that a major goal in the current paper is to understand the molecular basis of "copulation complementation" from the Xue and Noll paper. If this is the goal the paper succeeds 100%. But "choice" experiments would certainly increase interpretation of the evolutionary significance of these results. If "choice" experiments are not possible, I would at least recommend some discussion of how they might lead to additional insight.

We have added this to the Discussion, thank you for the suggestion.

2) If SP is a protein that binds sperm, is it that surprising that SP binds sperm from other males? At the outset, the null hypothesis seems to be that a protein that binds sperm will bind another male's sperm. What seems more interesting are the experiments that showed that the LTR-SFPS bind in sperm specific manner (subsection “Sex peptide binding to sperm of a prior male does not require receipt of LTR- SFPs from the second male” of Results, especially around the fourth paragraph). But this part of the paper is kind of tucked away, for example does not appear in the Abstract.

You are right that the LTR-SFPs result is very important, we were remiss failing to highlight it, especially in the Abstract. This has been remedied in the revised manuscript, particularly the Abstract and near the end of the Introduction. Thank you.

3) One could imagine that genetic relatedness among males is an important facet here. In the trivial case where the two male mates are genetically identical, SP would not be able to discriminate between self vs. non-self sperm. I realize that is not the experiment they did, and that in fact the two males in Figure 1 were derived from distinct strains. But my question is, how genetically distinct are they? Some more discussion of the genetic backgrounds is warranted. And what might happen if they used males that was more versus less genetically related? Might they see a difference in the sharing of SP? The reason I ask this question is because if this is some kind of "cooperation", it should be more apparent among related individuals. This might not be addressable, but might be useful to add to the Discussion.

The reviewer’s idea is excellent. We have added consideration of it to the Discussion, and appreciate the suggestion. It might be addressable in a large separate study, once very unrelated males with marked sperm are generated. This will be an interesting future direction, but is not possible at this time.